# Overview of Cellular Immunotherapies within Transfusion Medicine for the Treatment of Malignant Diseases

**DOI:** 10.3390/ijms22105120

**Published:** 2021-05-12

**Authors:** Nataša Tešić, Primož Poženel, Urban Švajger

**Affiliations:** 1Department for Therapeutic Services, Blood Transfusion Center of Slovenia, Šlajmerjeva 6, 1000 Ljubljana, Slovenia; natasa.tesic@ztm.si (N.T.); primoz.pozenel@ztm.si (P.P.); 2Faculty of Pharmacy, University of Ljubljana, Aškerčeva 7, 1000 Ljubljana, Slovenia

**Keywords:** transfusion medicine, cell-based immunotherapy, cancer

## Abstract

Over the years, transfusion medicine has developed into a broad, multidisciplinary field that covers different clinical patient services such as apheresis technology and the development of stem cell transplantation. Recently, the discipline has found a niche in development and production of advanced therapy medicinal products (ATMPs) for immunotherapy and regenerative medicine purposes. In clinical trials, cell-based immunotherapies have shown encouraging results in the treatment of multiple cancers and autoimmune diseases. However, there are many parameters such as safety, a high level of specificity, and long-lasting efficacy that still need to be optimized to maximize the potential of cell-based immunotherapies. Thus, only a few have gained FDA approval, while the majority of them are studied in the context of investigator-initiated trials (IITs), where modern, academically oriented transfusion centers can play an important role. In this review, we summarize existing and contemporary cellular immunotherapies, which are already a part of modern transfusion medicine or are likely to become so in the future.

## 1. Introduction

One of the main goals of transfusion medicine is to provide a safe and effective supply of blood products for the treatment or prevention of disease. Blood components, including red blood cells, fresh frozen plasma, and platelet concentrates are destined to go to patients affected by malignant and nonmalignant diseases, trauma, and sepsis, and for use in complex cardiac, orthopaedic and transplantation surgeries [1]. The collection, testing, processing, storage, and distribution of blood components to patients involves complex procedures supported by specific guidelines in order to ensure transfusion safety and efficacy [2]. The discovery of the ABO blood group system by Landsteiner overcame the serious problem of acute hemolysis allogenic blood transfusion. Additionally, further improvements in laboratory testing and donor screening reduced the risk of transfusion-transmitted infections and immunological reactions associated with transfusion [3,4].

Over the years, a segment of transfusion medicine has developed broad expertise in apheresis technology, cell and tissue cryobanking, quality management, and good manufacturing practice (GMP). In fact, transfusion services or blood establishments were the first institutions to deal with cell therapies for clinical use in the context of stem cell transplantation. This required thorough knowledge about hematopoietic stem cell collection, processing, and storage. A firm regulatory frame for the field has been derived from quality management systems overviewed by regulatory agencies already in place for blood-banking activity. All the ingredients for the next developing steps were at the blood establishments; therefore, it comes as no surprise that transfusion medicine in the last 15–20 years has made huge leaps forward toward novel cell therapies, immunotherapies, and regenerative medicine. Novel cell therapies developed in the field of transfusion medicine in the beginning relied on two major cell sources: bone marrow and peripheral blood.

In European Union regulations, gene and somatic cell therapies, as well as tissue engineering, are defined as advanced therapy medicinal products whose quality and efficacy must be demonstrated before their final approval for human and veterinary use. This includes cell therapies ranging from simple bone marrow transplantation, such as hematopoietic stem cell transplantation (HSCT), to complex tissue engineering [5]. Allogeneic HSCT for the treatment of hematological malignancies is internationally and historically the most established and the most frequently used mode of cell therapy. At the same time, it represents the first example of adoptive cell therapy (ACT). Hematopoietic stem cells are isolated from a genetically similar donor and transplanted into a patient whose bone marrow harboring disease has been eradicated by chemotherapy and/or irradiation [6]. Unlike unmanipulated allogeneic hematopoietic stem cells, immune cells intended for immunotherapy can be isolated, genetically modified, expanded, and used as vaccines against viruses as well as cancer.

Cancer is the second leading cause of death after cardiovascular diseases. In 2012, there were 14.1 million new cancer cases. According to global statistics, it is expected that the number of new cancer cases will increase to 21 million by 2030 (www.cancer.gov). For decades, surgery, chemotherapy, and radiotherapy have been the mainstay of cancer treatment. However, these conventional treatments have very limited efficacy in patients with late-stage or end-stage disease. In addition, disease relapse or progression is still a common problem due to residual malignant cells as well as tumor metastases. Nonetheless, the therapeutic efficacy of standard anticancer therapies can be enhanced by novel approaches that boost weakened immune systems [7].

To date, cellular immunotherapies have demonstrated objective responses even in late-stage disease after standard cancer treatments have failed [8]. However, an antitumor effect was achieved only in some patients with a select group of cancers while other patients did not respond to the therapy. Although this is the result of a great number of variables, the importance of patient-to-patient heterogeneity dictates future development of personalized immunotherapies.

## 2. Tumor Infiltrating Lymphocytes (TILs)

The term “tumor infiltrating lymphocytes” was coined in 1969 by Wallace Clark to describe the distinct part of a host’s response to cancer. The immune infiltrates are a heterogenous group of cells composed of effector T cells, natural killer (NK) cells, macrophages, B cells, dendritic cells (DCs), and other types of immune cells that interact most closely with the tumor cells [9]. Even though tumor-specific T cells are naturally present in cancer patients, they are relatively low in numbers, and their function is impaired. However, numerous studies have demonstrated that the presence of TILs in surgically resected tumor tissue was associated with a favorable clinical outcome in several cancers, including esophageal squamous cell carcinoma [10], high-grade serous ovarian cancer [11], and non-small-cell lung cancer [12]. The most consistent positive prognostic value amongst TILs was demonstrated for T cells, especially cytotoxic T lymphocytes (CTLs). Namely, it is well known that CTLs play a key role in anticancer immunity by direct killing of malignant cells. However, tumors possess many mechanisms to resist destruction by CTLs such as altering chemokine expression, thereby reducing T-cell infiltration into the tumor site, or by suppressing their function via activation of T-cell inhibitory (checkpoint) pathways leading to inhibition of T-cell activation pathways and suppression of NK-cell activity [13,14].

Due to the suppressive tumor environment (TME), T cells within tumor are often in low numbers and incapable of controlling tumor growth [15]. However, development of protocols using interleukin 2 (IL-2) has had significant impact on development of cancer immunotherapies. Because IL-2 is a potent T-cell growth factor, it has been widely used for expansion of T cells intended for both in vitro analysis and immunotherapy protocols. The generation of antitumor T cells is based on the extraction of TILs from surgically resected tumor fragments and ex vivo expansion with high doses of IL-2 (Figure 1). These highly activated TILs are then infused back into the patient following lymphodepleting chemotherapy. In addition, after T-cell infusion patients are treated with high doses of IL-2 with the purpose of promoting T-cell activation and proliferation in vivo [16,17]. TIL-based ACT in combination with high doses of IL-2 has been widely studied in patients with metastatic melanoma and has showed significant improvement in clinical outcomes [18,19]. Furthermore, TIL therapy has now been explored in cancers other than melanoma, such as ovarian, cervical, kidney, and uterine cancers [20,21]. However, the administration of high doses of IL-2 together with TILs can induce, although transient, severe toxicities requiring intensive medical care and patient monitoring. Furthermore, clinical studies have demonstrated that in vivo expansion of Tregs can occur when treating patients either with low-dose IL-2, e.g., after HSCT [22], or when using high IL-2 doses as was shown in melanoma patients [23].

Because high doses of IL-2 can induce severe side effects, new clinical trials have started to explore whether lower doses of IL-2 can achieve an equivalent therapeutic response and overcome the previously mentioned disadvantages. Clinical study results obtained in phase I/II trials published by Andresen et al. show that long-lasting complete responses in patients with metastatic melanoma was achieved after TIL-based ACT with low doses of IL-2 [24]. In addition, a nonrandomized phase I trial was recently completed where patients with metastatic melanoma and ovarian cancer were treated in order to examine safety and efficacy of TIL-based ACT with low doses of IL-2 (ClinicalTrials.gov Identifier: NCT03158935).

## 3. Genetically Modified T Lymphocytes

T cells can be engineered to express either antigen-specific T-cell receptors (TCRs) or chimeric antigen receptors (CARs) (Figure 2). The generation of effective tumor-specific TCR-T cells requires an identification of an appropriate target sequence which can be cloned from rare naturally occurring tumor-reactive T cells, isolated from cancer patients. Other valuable resources for the isolation of high-avidity TCRs against human antigens are transgenic mice that express human leukocyte antigen (HLA) and can be immunized with the target tumor antigens. After immunization, genes encoding the new TCR α and β chain are isolated from mouse T cells that are specific for the target antigen and cloned into retroviral vectors [25]. This approach has been successfully used for generation of modified autologous T cells that have been in vitro expanded with high-avidity TCRs against melanoma antigens [26] and human carcinoembryonic antigen (CEA) [27]. Modified TCR-T cells recognize tumor-associated antigens (TAAs), which are presented by antigen presenting cells (APC) in an HLA-dependent manner. Unlike TCR-T cells, T cells carrying CARs can recognize surface specific antigens and eliminate tumor cells in an HLA-independent manner, which can render more tumor cells susceptible to their attack [28]. CAR is a hybrid molecule typically composed of a single-chain variable fragment (scFv) of a tumor targeting domains fused to T-cell signaling and costimulatory receptors [29].

Development of CAR-T cells has progressed over the last two decades. First-generation CAR-T cells used the signal transduction domain of a CD3 protein for CAR construction. However, their efficacy was limited by a lack of costimulation. In order to strengthen and prolong T-cell activation, second-generation CAR-T cells were modified with an additional costimulatory domain, either CD28 or 4-1BB. Third-generation CAR-T cells were engineered with the combination of two costimulatory domains, the first domain being either CD28 or 4-1BB, and the second domain being CD28, 4-1BB, or OX40. In the past few years, fourth-generation CAR-T cells, so-called “TRUCKs”, have been additionally modified to express cytokines such as IL-12, whose expression is under the control of a constitutive or inducible promoter. This helps promote T-cell activation and helps in the recruitment of native immune cells to eliminate tumor cells that have not been recognized by CAR-T cells [30].

### 3.1. CAR-T Cell-Based Immunotherapy

Over the last decade, CAR-modified T-cell therapy has progressed rapidly due to the development of synthetic biology and gene therapy and because of the potential these cells may have in the treatment of malignant diseases. CAR-T-cell immunotherapy has been predominantly used in the treatment of hematological diseases, such as relapsed/refractory (R/R) B cell acute lymphoblastic leukemia (B-ALL), B-cell non-Hodgkin’s lymphoma (B-NHL), and chronic lymphocytic leukemia (CLL). Currently, the most studied and rewarding target of CAR-T-cell therapy is the B-cell specific antigen CD19. Response rates ranging from 70% to 94% have been achieved in several clinical trials [31]. This led the Food and Drug Administration (FDA) to approve two second generation anti-CD19-directed genetically modified autologous CAR-T-cell therapies, Kymriah^®^ (tisagenlecleucel) and Yescarta^®^ (axicabtagene ciloleucel). Kymriah^®^ is indicated in the treatment of both pediatric patients and young adults up to 25 years of age with R/R B-ALL, as well as in the treatment of adults with R/R diffuse large B-cell lymphoma (DLBCL). Yescarta^®^ was approved in the treatment of adults with R/R large B-cell lymphoma (www.fda.gov). More recently, the FDA approved three additional CAR-T therapies, namely brexucabtagene autoleucel (Tecartus^®^), lisocabtagene maraleucel (Breyanzi^®^), and idecabtagene vicleucel (Abecma^®^). Abecma^®^ is the most recently approved CAR-T-cell therapy directed toward B-cell-maturation antigen (BCMA). It is indicated for the treatment of adult patients with R/R multiple myeloma. Breyanzi^®^ was also approved in 2021 and targets CD19. It was approved for the treatment of DLBCL, high-grade B-cell lymphoma, primary mediastinal large B-cell lymphoma and follicular lymphoma grade 3B. Tecartus^®^ was approved in 2020 for the treatment of R/R mantle cell lymphoma, and the results were published by Wang and colleagues [32].

The successful and efficient eradication of malignant cells with anti-CD19 CAR-T cells is accompanied with increased incidence of serious side effects, compared to other cellular immunotherapies. Namely, the intensity of the immunological response can result in the development of cytokine release syndrome (CRS) with the risk of multiple organ failure. Additionally, every treatment can result in tissue damage due to the unwanted expression of a target antigen in normal tissues or organs especially in lymphatic tissues. Although CD19 is an ideal target for CAR-T-cell immunotherapy, the success rate of CARs can be unsatisfactory because of antigen loss thereby requiring the discovery of additional targets. CARs need to be further optimized to improve T-cell activation, antitumor activity, and recognition specificity, as well as safety. Currently, other B-cell specific antigens are under investigation as potential targets, for example CD20, CD22, CD30, CD38, and CD138 [33]. From the novel targets, CD20 and CD22 are those most frequently studied in clinical trials.

Whereas remarkable results have been achieved by the CAR-T-cell treatment of patients with B-cell haematological malignancies, less-promising conclusions can be derived from clinical trials using CAR-T cells designed to treat solid tumors. A lack of suitable target antigens, inefficient CAR-T cell trafficking into the tumor, and the TME have emerged as the major problem areas that limit the efficacy of CAR-T cells in the treatment of solid tumors [34]. Therefore, much attention has been paid to the identification of TAAs, as well as the improvement of CAR-T cell trafficking in the treatment of solid tumors. The targeting of TAAs with CAR-T cells has shown a significant antitumor effect in numerous in vitro as well as in vivo studies, and many of these therapies have been translated into clinical trials for patients with solid tumors. According to the ClinicalTrials.gov database, the most studied targets for solid tumors are the epidermal growth factor receptor (EGFR), the human epidermal growth factor receptor 2 (HER2), mesothelin (MSLN), cell membrane mucin-1 (MUC1), and the prostate specific membrane antigen (PSMA).

EGFR variant III (EGFRvIII) is the most common variant of the EGFR observed in human tumors and is expressed in about 30% of newly diagnosed glioblastomas. Current results obtained from phase 1 clinical trial (NCT02209376) indicate that the manufacture and intravenous infusion of EGFRvIII CAR-T cells is feasible and safe without the appearance of cytokine release syndrome. Additionally, the detection of EGFRvIII CAR-T cells in high concentrations in the brains of two patients, both with tumor resection within 2 weeks post EGFRvIII CAR-T infusion, suggests effective trafficking and considerable expansion of these cells within active regions of glioblastoma [35]. MSLN is a cell-surface antigen which is an attractive CAR target because of its low expression in normal mesothelial cells and high expression in a wide range of solid tumors such as mesothelioma, breast, lung, pancreas, ovarian, and other tumors. Several preclinical studies targeting tumors with high MSLN expression have shown promising potential of this molecule as a target. Therefore, multiple phase I clinical trials have been initiated with the aim of evaluating safe doses and the maximum tolerated dose of MSLN CAR-T cells [36]. Furthermore, Beatty and colleagues reported on the safety, feasibility, and antitumor effects of MSLN specific mRNA CAR T-cells after repetitive infusions in patients with malignant pleural mesotheliomas and metastatic pancreatic cancer [37]. However, the efficacy of MSLN CAR-T-cell therapy can be limited due to the overexpression of inhibitory molecules in the tumor environment, such as the programmed death-ligand 1 (PD-L1). To overcome this obstacle, MSLN CAR-T-cell therapy is combined with anti-PD-1/PD-L1 monoclonal antibodies or clustered regularly interspaced short palindromic repeats (CRISPR) associated protein 9 (Cas9) to knock out the PD-1 of the CAR-T cells. Clinical trials with PD-1 knocked-out mesothelin-directed CAR-T cells are underway for various solid tumors (NCT03747965, NCT03545815).

The safety and efficacy of HER2 CAR T-cells in patients with R/R HER2-positive sarcoma, pancreatic cancer, and glioblastoma have been evaluated in several clinical studies. Detection of cytomegalovirus (CMV) antigens, such as pp65 in glioblastoma tissue indicate CMV as a contributing factor to glioblastoma progression and have suggested using immune-based therapies as a new target [38]. Ahmed et al. performed a phase 1 dose-escalation study investigating autologous T cells genetically modified to express HER2-CARs in cytomegalovirus (CMV)-seropositive patients with progressive HER2-positive glioblastoma. Among 17 subjects, eight experienced a clinical benefit, with the median overall survival rate being 11.1 months from the time of first infusion and 24.5 months after diagnosis [39], while the median survival of glioblastoma patients was 12.1 months after treatment with radiotherapy [40].

However, there is currently no CAR-T-cell therapy able to induce a consistent and lasting regression in solid tumors.

### 3.2. TCR-Based Immunotherapy

The major advantage of TCR-based immunotherapies is their ability to recognize the disease-associated intracellular proteins presented by HLA molecules as peptide fragments. This mechanism of antigen recognition enables T cells to eliminate not only virus-infected cells but also tumor cells. It is known that T cells are tolerant to self-antigens derived from self-proteins; however, point mutations in tumor cells may create novel HLA-binding residues that can elicit a robust T-cell response [41].

It was first reported that the adoptive transfer of TCR-modified T cells (TCR-T) recognizing melanocyte-specific differentiation antigen 1 (MART-1) could mediate objective cancer regression in patients with progressive metastatic melanoma [42]. Encouraging results enabled the initiation of new clinical trials which started to examine different tumor antigen targets, such as melanoma-associated antigens (MAGE-A) and New York esophageal squamous cell carcinoma 1 (NY-ESO-1). Robbins et al. reported a clinical response in melanoma patients (55%) and synovial cell sarcoma patients (61%) after the adoptive transfer of autologous TCR-T cells recognizing HLA-A*0201-restricted NY-ESO-1 epitope [43]. However, some patients developed severe toxicities after the adoptive transfer of TCR-T cells, such as respiratory distress and mental disturbances. Chodon et al. have reported that two melanoma patients developed respiratory distress after treatment with ACT using autologous MART-1-specific TCR-T cells. The level of circulating cytokines and chemokines was increased and comparable to those observed in acute pneumonia [44]. Furthermore, treatment of three patients with autologous anti-MAGE-A3 TCR-T resulted in neurologic toxicity probably due to the recognition of MART-A12 protein expressed in a subset of neurons in the brain [45]. During the manufacture of TCR-T cells, the mismatch between the newly introduced TCR receptor and the endogenous TCRs in T cells can negatively affect the expression of transduced TCR. Recently, Sun and colleagues demonstrated a successful generation of MAGE-A4-restricted T cells with a silenced endogenous TCR using small interfering RNA. The effects of such TCR-T cells were tested in a preclinical model and were successfully used in a single case, uterine leiomyosarcoma patient [46].

## 4. Other Adoptive T-Cell Therapies

### 4.1. γ/δT Cell-Based Immunotherapy

γ/δT cells represent a small (less than 5%) group of heterogeneous T cells that serve an important role in infectious diseases as well as various types of cancer. Unlike conventional αβ T cells, their TCR contains δ and γ chains. γ/δT cells can destroy target cells directly by their cytotoxic activity or indirectly through the activation of other immune cells. Furthermore, they can also act as APCs. Owing to these features, γ/δT cells have received increasing attention in recent years [47].

The major subset of γ/δT cells in peripheral blood are Vγ9δ2 T cells with antitumor and antiviral defense abilities. Furthermore, Vγ9δ2 T cells have been considered as potential candidates for immunotherapy because they recognize antigens in an MHC-independent manner and therefore should not induce graft-versus-host disease (GvHD) [48]. They can be obtained from peripheral blood mononuclear cells (PBMC) and expanded in vitro by cytokines (e.g., IL-2) and antigens (e.g., isopentenyl pyrophosphate). In addition, Vγ9δ2 T cells can also be expanded in vivo by phosphomonoester antigen.

Wilhiem and colleagues were the first to demonstrate that the in vivo activation of Vγ9δ2 T cells in patients with lymphoid malignancies by Pamidronate and low-dose IL-2 was safe and induced a clinical response [49]. In addition, Zoledronate-induced in vivo expansion of Vγ9δ2 T cells together with a low dose of IL-2 was also well tolerated and demonstrated an antitumor effect in the treatment of patients with prostate cancer [50] as well as neuroblastoma [51]. Furthermore, the adoptive transfer of ex vivo expanded Vγ9δ2 T in combination with Zoledronate given intravenously was demonstrated to be safe and feasible and produced a clinical response in patients with metastatic solid tumors [52], advanced hematological malignancies [53], and gastric cancer [54]. Recently, four phase I/II clinical trials have been completed with the aim to investigate the safety and efficacy of adoptively transferred γ/δT cells in combination with tumor-reducing surgery against breast (NCT03183206), lung (NCT03183232), liver (NCT03183219), and pancreatic cancer (NCT03180437).

In recent years, researchers have started to explore the efficacy of γ/δT cells as CAR carriers. A preclinical study carried out by Capsomidis et al. showed for the first time that γδT cells transduced with a second generation of disialoganglioside (GD-2)-specific CAR are capable of migrating toward neuroblastoma cells and have potent tumor antigen-dependent cytotoxicity in vitro [55]. In September 2019, a phase I dose-escalation clinical study was initiated to investigate safety and tolerability of haploidentical, or allogeneic γδT cells modified with natural-killer group 2 member D ligand (NKG2DL)-specific CAR in subjects with relapsed or refractory solid tumors (NCT04107142).

### 4.2. Cytotoxic T-Cell (CTL)-Based Immunotherapy

The adoptive transfer of ex vivo-generated antigen-specific CTLs is a promising approach in cellular immunotherapy and has been studied in the treatment of certain infectious diseases and cancers such as metastatic melanoma [56]. CTLs have a crucial role in host defense against intracellular pathogens and tumor surveillance. The primary advantage of using CD8+ T cells for adoptive T-cell cancer therapy is their ability to specifically target tumor cells through the recognition of differentially expressed tumor antigens. Upon stimulation, naïve CD8+ T cells undergo antigen-driven generation of effector CTLs, which destroy target cells through the induction of apoptosis.

One of the main hurdles of adoptive cell therapy is generating enough functional antigen-specific T cells. Initial ex vivo CTLs expansion included repeated stimulation of autologous peripheral blood or TIL-derived CD8+ T cells with autologous DCs pulsed with tumor antigens. Additionally, other less potent forms of APCs have been used to induce antigen-specific CTLs, such as CD40-activated B cells [57]. However, although DCs are the most efficient APCs for T-cell activation, DC-based T-cell expansion is challenging. Their isolation, enrichment, and expansion are time-consuming, technically difficult, and costly procedures. Furthermore, the limited replicative potential of DCs is an obstacle to their use in large-scale adoptive T-cell therapies. Therefore, researchers developed artificial APCs (aAPCs) to generate T-cell grafts for effective adoptive T-cell cancer therapy [58].

In the case of cell-based aAPCs, the human erythroleukemic cell line K562 has been widely utilized as the cell backbone for a series of aAPCs. In addition to endogenous HLA class I molecules, costimulatory molecules were also transduced to the aAPCs [59]. An example of how K562-derived aAPCs can generate antigen-specific CTLs was published in a study by Butler and colleagues. They demonstrated that aAPCs expressing HLA-A2, CD80, and CD83 were able to expand and maintain long-lived antitumor-specific CD8+ T cells for adoptive immunotherapy [60]. However, although K562-aAPCs cells are irradiated before they are used to expand T cells, their malignant origin is an important drawback considering their infusion in cancer patients. This prompted the development of the bead-based approach for generating large numbers of antigen-specific CTLs. It was demonstrated that magnetic beads coupled to HLA-Ig dimer and anti-CD28 specific antibody induce and expand CTLs specific for CMV or MART-1 [61]. Furthermore, Perica and colleagues developed a novel enrichment and expansion protocol using paramagnetic nanoscale aAPCs to rapidly expand tumor-specific T cells from rare naïve precursors [62]. Nano-aAPCs are synthesized by coupling chimeric HLA-Ig dimer and an anti-CD28 antibody to 50–100 nm paramagnetic iron-dextran nanoparticles. In addition, Hickey et al. have prepared aAPCs from superparamagnetic iron oxide nanoparticles (SPIONs) that can be used to mimic receptor clustering via binding magnetic particles on the surface of T cells by applying a magnetic field. They have demonstrated that aAPCs which are prepared from SPIONs and are larger than 300 nm are more effective at activating CD8+ T cells than smaller, 50 nm aAPCs [63]. Further investigations are necessary to develop optimal aAPC protocols to generate functional antigen-specific T cells for broad use in clinics.

In light of the current epidemic, there are also efforts to battle COVID-19 using antigen-specific T cells. Although there are no published reports up to date, there are several registered clinical trials, many of them actively recruiting patients. As can be seen from the database Clinicaltrials.org, these studies are active all over the world, from the United States to Europe to Asia. They can be found under identifier numbers NCT04457726 (Singapore), NCT04762186 (Germany), and NCT04401410 (Houston, TX, USA), to name just a few. The generation of SARS-CoV-2-specific T cells was demonstrated to be feasible using automatic enrichment devices, such as CliniMACS Prodigy^®^ [64]. Furthermore, the increasing number of convalescent patients will facilitate donor recruitment in the future.

### 4.3. Donor Lymphocyte Infusion (DLI)-Based Immunotherapy

Allogeneic HSCT (allo-HSCT) is an established therapy in the treatment of patients with hematologic malignancies. Unfortunately, relapse remains the main cause of treatment failure with a poor prognosis. Formerly, a second allo-HSCT was the predominant strategy for patients with relapsed or refractory disease but had limited success and a high mortality rate. One of the effective alternative therapies to treat or prevent tumor relapse posttransplant is DLI, also known as buffy coat infusion or donor leukocyte infusion. The main goal of DLI is to induce durable remissions by enhancing the graft-versus-leukemia (GVL) effect which is primarily mediated by donor T cells. However, the most common treatment-related side effect after unmodified DLI is GvHD (40–60%).

DLI was applied in the treatment of various hematological malignancies. The best and most durable responses were seen in patients with relapsed chronic myeloid leukemia (CML). In 1990, Kolb and colleagues provided the first evidence that DLI can induce remission of a disease following a disease relapse [65]. Overall, DLI induced durable responses in more than 70% of the patients with relapsed chronic phase CML. Furthermore, response rates in patients with cytogenetic and chronic phase relapse were higher than in patients in an accelerated phase or blast crisis [66]. While an allogeneic GVL effect of DLI is most pronounced in patients with chronic CML, the response rate and durability appear lower in patients with multiple myeloma and acute leukemia, particularly in those with ALL. The European Bone Marrow Transplant (EBMT) group retrospectively reviewed the outcomes of 399 patients with acute myeloid leukemia (AML) who had been treated with or without DLI. Their results confirmed that a clinical benefit of DLI on AML relapse after allo-HSCT was limited to a minority of patients [67]. Therefore, new strategies were needed to enhance the therapeutic efficacy of DLI for relapsed hematologic malignancies. Nikiforow and colleagues reported results from a phase I clinical trial using CD25/Treg-depleted DLI infusions in patients with hematologic malignancies who had relapsed after transplantation. These modified DLIs were well-tolerated and associated with a better response rate compared to unmodified DLIs [68]. Furthermore, the combination of DLI with the FDA-approved drug blinatumomab, a bi-specific constructed Ab targeting CD19 and CD3, was also investigated in clinical trials. The first report was published in 2016 for the treatment of relapsed B-ALL after allo-HSCT. Recently, Durer et al. have described the case of a 51 year-old patient with relapsed mixed phenotype acute leukemia (MPAL) treated with the combination therapy of blinatumomab and DLI. The combined therapy resulted in a longer period of complete remission compared to allo-HSCT with chemotherapy [69]. However, additional clinical studies are needed to determine the safety of this promising therapy. In June 2019, a phase II study was designed to evaluate safety and tolerability, as well as the duration of the response after the combined treatment of DLI and blinatumomab in B-ALL patients (NCT03982992). Recently published results by Park et al. have demonstrated that CTLA4-CD28 chimera gene modification (CTC28) of CD4 and CD8 T cells significantly increased the GVL effect of DLI in a murine model of ALL [70]. Because disease response to DLI and presence of GvHD are strongly correlated, different approaches are focused to reduce GvHD without impairing GVL effect. Administration of donor lymphocytes modified by herpes simplex virus thymidine kinase suicide gene (HSV-TK) was confirmed to be safe and efficient and to reduce GvHD in patients with relapsed hematologic malignancies after allo-HSCT [71]. Additionally, DLI with tumor-specific donor lymphocytes is also an effective approach to separate GvHD from GVL; however, the major limitation is the lack of TAAs [72].

## 5. NK Cell-Based Immunotherapy

Although T-cell-based immunotherapies have been demonstrated to possess efficiency in the treatment of cancer patients, their accompanying disadvantages have led to the need to seek other immunotherapy approaches, particularly with the aim to reduce the GvHD effect. NK cells, the key components in the innate immune system, have a great potential for cancer immunotherapy as they normally do not induce GvHD and are able to eliminate tumor cells or virally infected cells without prior stimulation. Nevertheless, tumors possess many mechanisms which lead to the dysfunction of NK cells; therefore, restoring the impaired antitumor activity of NK cells in cancer patients is a fundamental therapeutic objective.

### 5.1. Adoptive Transfer of Unmodified NK Cells

The initial treatment involved the adoptive transfer of IL-2-activated autologous NK cells derived from PBMC. Clinical studies demonstrated that transfer of ex vivo IL-2 expanded autologous NK cells alone or in combination with subcutaneous administration of IL-2 was safe and feasible in the treatment of solid tumors as well as hematological diseases [73]. However, although the adoptive transfer of autologous NK cells resulted in an increased number of circulating NK cells in peripheral blood, no improvements in disease outcomes were observed [74,75]. The main cause of inefficacy was the matching between the self-HLA molecules of tumor cells and killer cell immunoglobulin-like receptors (KIR) on autologous NK cells, resulting in NK-cell inhibition. This obstacle forced researchers to investigate the possibility of NK-cell allograft as an adoptive treatment for cancer.

The first evidence of NK-cell clinical benefit was reported two decades ago in high-risk AML patients after HLA mismatch donor hematopoietic transplantation [76]. Clinical trial results showed that NK-cell alloreactivity, which was triggered by KIR ligand incompatibility, resulted in a GVL effect with no development of GvHD. Later, numerous clinical trials were initiated with the aim of investigating alloreactive PBMC-derived NK cells as an immunotherapy approach in the treatment of hematological disorders as well as solid tumors. Results obtained from different studies confirmed a therapeutic effect in patients with high-risk myelodysplastic syndromes (MDS) and AML after infusion of haploidentical NK cells. Furthermore, NK-cell infusions were safe and well tolerated without GvHD [31,77]. For the treatment of solid tumors, adoptive NK-cell therapy is often combined with tumor-specific monoclonal antibodies, which act by promoting NK-cell antibody-dependent cell-mediated cytotoxicity (ADCC) through the binding of the IgG Fc region and its activating Fc receptor (FcR) expressed on NK cell. Unlike other effector cells, NK-cells harbor only activating FcγRs (FcγR IIIa, also known as CD16a, and FcγR IIc, also known as CD32c), and therefore, these cells are considered the most important effectors for inducing ADCC [78]. Recently, Ishikawa et al. have demonstrated that infusion of allogeneic NK cells in combination with trastuzumab and cetuximab resulted in a reduction of tumor size in patients with advanced gastric or colorectal cancer [79]. Additional administration of IL-2 is frequently associated with NK-cell therapy to induce their in vivo expansion; however, IL-2 can attenuate their efficacy by simultaneously causing the expansion of Tregs [80]. In more recent years, IL-15 has been emerging as a promising substitute for IL-2. For example, in a study by Colley et al., they demonstrated a successful treatment of advanced acute myeloid leukemia patients with recombinant human IL-15 and haploidentical NK cells [81].

The most common source of allogeneic NK cells are donor PBMCs, but they can be also obtained from umbilical cord blood (UCB) and bone marrow. Additionally, induced pluripotent stem cells (iPSCs) and human embryonic stem cells (hESCs) represent valuable source of NK cells [82]. In 2016, Michel et al. published the results they obtained in a phase III study of single-versus-double UCB-derived NK-cell transplantation in pediatric patients with hematologic malignancies. Clinical trial results showed that approximately 70% of patients were disease-free two years after either single or double UCB-derived NK transplantation [83]. Currently, there are six clinical trials (NCT03420963, NCT03634501, NCT02727803, NCT01619761, NCT02722668, and NCT03019640) recruiting patients with hematological or solid cancers in order to evaluate the safety, highest tolerable doses, and effectiveness of UCB-derived NK cells alone or in combination with chemotherapy.

Another way to generate a large number of NK cells is through a clonal NK cell line. There are several established cell lines among which NK-92 has shown the highest antitumor cytotoxicity. Furthermore, NK-92 is not only the most used but has also received FDA approval for clinical applications. This is an IL-2 dependent NK cell line, originating in patients with NHL, which requires their irradiation before infusion to the patient to avoid allogeneic tumor engraftment [84]. Several clinical trials confirmed the safety and feasibility of the NK-92 cell line in patients with hematological diseases. However, the therapeutic efficacies of NK-92 remain limited, mainly due to their lack of CD16 expression and consequent inability to participate in ADCC [85,86].

### 5.2. Genetically Modified NK Cells

A variant of the NK-92 cell line, the high-affinity NK cell line (haNK), was modified to express CD16A receptor and endogenous IL-2 [87]. In February 2019, a phase II clinical trial was initiated in order to evaluate the efficacy of haNK in combination with monoclonal antibody Avelumab in patients with advanced Merkel Cell Carcinoma (NCT03853317). In addition, promising results obtained in clinical trials using CAR-T cells motivated researchers to modify NK cells with CARs. Namely, CAR-NK cells have some advantages over CAR-T cells including their isolation from multiple sources, lysis of tumor cells through both a CAR-dependent and CAR-independent manner, and have limited persistence in circulation, leading to few on-target/off-tumor side effects. Furthermore, CAR-NK cells may also reduce risk of inducing severe CRS. Namely, activated NK cells usually produce interferon γ (IFN-γ) and granulocyte–macrophage colony-stimulating factor (GM-CSF), whereas the CRS induced by CAR-T cells is mainly mediated by pro-inflammatory cytokines, such as IL-6, IL-1, and tumor necrosis factor α (TNF-α) [88]. Additionally, infusion of allogeneic NK cells obtained from HLA-matched or haploidentical donors do not cause GvHD in patients with cancer [89].

Preclinical studies using CAR-NK cells demonstrated encouraging results against different types of tumor antigens in both hematological diseases and solid tumors [90,91]. Subsequently, the results obtained enabled the initiation of clinical trials with CAR-NK-cell therapy. At present, two phase I clinical trials, with the status of “not yet recruiting”, are focusing on an evaluation of the safety and efficacy of anti-CD19 (NCT03690310) and anti-CD22 (NCT03692767) CAR-NK cells in patients with relapsed refractory B-cell lymphoma. Additionally, the safety and feasibility of anti-PSMA CAR-NK cells will be investigated under a phase I clinical trial on patients with prostate cancer (NCT03692663). Studies of CAR-NK cells already recruiting patients under phase I or phase II can be found for NK-CARs targeting HER2 for glioblastoma/gliosarcoma therapy (NCT03383978), CD19 for R/R B-cell-related malignancies (NCT03056339), the inhibitory ligand PD-L1 for treatment of advanced solid tumors (NCT04050709), or pancreatic cancer in combination with a biological drug N-803 (NCT04390399), which is an IL-15R superagonist [92].

NK-92 or UCB-derived NK cells have also been genetically modified in order to obtain NK cells with stable CAR-expression. To date, fewer clinical studies have been initiated using CAR-NK-92 cells. However, a phase I/II clinical trial, in the recruiting phase, will involve patients with relapsed and refractory multiple myeloma who will be treated with B-cell maturation antigen (BCMA) CAR-NK-92 cells to assess the safety and feasibility of this strategy (NCT03940833).

## 6. Cytokine-Induced Killer Cell (CIK)-Based Immunotherapy

CIK cells are heterogeneous populations of effector CD3+CD56+ natural-killer T cells that can be obtained from PBMC, UCB, and bone marrow, followed by in vitro expansion with IFN-γ, anti-CD3 antibody, and IL-2. Because CIKs contain different cell subpopulations and have a much broader antitumor spectrum due to their non-MHC-restricted cytotoxicity, they have become one of the promising cell types for cancer immunotherapy. Extensive research work has been done with the aim of improving the safety and therapeutic response to CIK cell therapy. Namely, various clinical studies started to combine CIKs with chemotherapy, additional cytokines, DCs, antibodies, and immune checkpoint inhibitors [93]. Up until today, 121 clinical trials have been registered in the National Institutes of Health (NIH) Clinical Trial Data Bank using CIKs for the treatment of hematological malignancies as well as solid tumors.

The first report of the efficient and safe clinical use of IL-2-transfected autologous CIKs was demonstrated in patients with metastatic renal cancer, colorectal cancer, and lymphoma more than two decades ago [94]. The safety and clinical efficacy of autologous CIK cell immunotherapy combined with standard chemotherapy regimens was reported in patients with lung cancer after surgery [95] and triple-negative breast cancer [96]. Additionally, autologous DCs combined with CIKs followed by chemotherapy could prolong overall survival in patients with advanced colorectal cancer [97]. An important approach to redirect CIKs to target tumors is their genetic modification with CARs. Preclinical studies using CAR-modified CIKs demonstrated in vitro antitumor efficacy against different targets, such as EGFR [98] and ErbB2 [99]. The results obtained have boosted scientific interest such that clinical trials using CAR-modified CIKs have been initiated. In a recruiting phase I/II clinical trial, the recommended dose and safety of anti-CD19 CAR-CIK will be evaluated in adult and pediatric patients with B-ALL (NCT03389035).

## 7. Dendritic Cell (DC)-Based Immunotherapy

The identification of a novel cell type among mouse splenocytes by Nobel Laureate Ralph Steinman in the early 1970s opened up a new era in immunotherapy [100]. Through years of investigation, DCs were found to be the most potent APCs with the unique ability to initiate and maintain an immune response which can be immunogenic or tolerogenic. Dendritic cells can be categorized into several types with differences in morphology and function, depending on their origin and localization. In general, they include plasmacytoid (pDCs), conventional/myeloid (cDCs), and Langerhans DCs which are the most critical and effective DCs. In addition, monocytes can also differentiate into inflammatory dendritic cells (moDCs) during inflammation [101].

DCs represent an important link in the interface between the innate and adaptive immune systems. They have a key role in the activation and proliferation of naive CD8+ and CD4+ T lymphocytes by presenting antigenic peptides via major histocompatibility complexes (MHC) class I and II, respectively. In addition, DCs also express costimulatory molecules, such as CD80 and CD86, which provide a second signal to T cells via CD28 molecule on their surface. During antigen presentation DCs can secrete large quantities of cytokines, such as IL-12 p70, which direct polarization of CD4+ lymphocytes into T helper type 1 (Th1) cells. Furthermore, secreted cytokines also activate other immune cells and attract them to the site of infection [102].

Due to their potent antigen presentation and T-cell activation, DCs have generated a great deal of interest in their use as vaccines for cancer, as well as for infectious diseases. The safety and efficacy of DC-based vaccines have been investigated for decades in more than 400 registered clinical trials. DCs represent 0.5–1% of the leukocyte population in PBMC, meaning the isolation of the sufficient cell numbers necessary for a therapeutic dose is limited. To overcome this obstacle, DC-based vaccines can be prepared ex vivo as well as in vivo. In the ex vivo approach, large number of DCs are obtained by the in vitro culture of monocytes from peripheral blood [103] or CD34+ progenitor cells [104] in the presence of GM-CSF with cytokines such as IL-4 and TNF-α, respectively. The ex vivo generation of DC-based vaccines also requires the selection of proper tumor antigens and appropriate techniques for loading DCs with tumor antigens. Different methods have been used to load TAAs into DCs including transfection, fusion of DCs and tumors cells, and pulsing with synthetic peptides or purified proteins, DNA, and tumor lysate (Figure 2). Another important strategy in DC-based vaccine design includes determining the optimal route of vaccine administration in order to ensure the migration of loaded DCs to lymphoid organs where they can stimulate efficient T-cell responses [105]. However, the ex vivo preparation of DC-vaccines has some disadvantages, such as high costs and the time needed for laboratory procedures. On the other hand, DCs can be activated in vivo using specific peptides combined with GM-CSF or genetically modified cancer cells expressing GM-CSF [106].

DC-based immunotherapy has been tested in clinical trials in patients with a wide variety of cancers, such as malignant melanoma, prostate cancer, B-cell lymphoma, multiple myeloma, and hepatocellular carcinoma. More than 20 years ago, Mukherji et al. published the first clinical report of DC vaccinations performed on patients with advanced melanoma [107]. Since then, numerous clinical studies in melanoma patients have been performed using moDCs loaded with MAGE-, MART-, gp100-, or tyrosinase-derived peptides. Despite the fact that DC vaccination induced tumor specific T-cell responses in patients with advanced melanoma, clinical responses were limited mainly because of immunosuppression mechanisms in the tumor microenvironment. However, the clinical efficacy of DC vaccination in advanced melanoma patients might be enhanced by combining it with therapies that target these mechanisms. One of the approaches is the depletion of Tregs, major players in the tumor microenvironment that suppress effective antitumor response. In 2010, Jacobs et al. showed that the treatment of melanoma patients with anti-CD25 antibody daclizumab induced transient depletion of Tregs from the peripheral circulation. However, the efficacy of the DC vaccine was not enhanced, probably because daclizumab also destroys cytotoxic CD8+ T cells expressing CD25 [108]. Considerable progress was made by combining DC vaccination with immune checkpoint inhibitors that block cytotoxic T-lymphocyte-associated antigen 4 (CTLA-4). In a phase II clinical study, advanced melanoma patients were treated with ipilimumab combined with autologous moDCs electroporated with TriMix-mRNA (CD40L-, CD70-, and caTLR4-encoding mRNA) and mRNA encoding one of four melanoma-associated antigens (MAGE-A3, MAGE-C2, tyrosinase, or gp100) fused to an HLA-class II targeting signal. Clinical trial results showed tolerability and an overall tumor response rate of 38% in 39 metastatic melanoma patients [109]. More recently, the efficacy of combination of TriMix DC vaccine with ipilimumab was also confirmed in terms of a robust cytotoxic immune response in peripheral blood of melanoma patients [110]. There are several ongoing clinical trials evaluating the safety and tolerability of a peptide-loaded autologous DC vaccine in patients with stage III and stage IV melanoma (NCT03092453; NCT03325101), non-Hodgkin lymphoma (NCT03035331), and recurrent glioblastoma (NCT04201873) followed by treatment with the clinically approved anti-PD1 antibody pembrolizumab. A group from Osaka, Japan, reported on a single case of a patient with recurrent primary central nervous system lymphoma [111]. After several unsuccessful chemotherapy and radiotherapy treatments, the patient received nivolumab in combination with DC vaccine treatment. Remission was achieved after six cycles of nivolumab. A more recently published proof-of-concept trial further explored a combinatorial approach wherein three patients with resected pancreatic adenocarcinoma were treated with DC vaccine loaded with neoantigens [112]. The future study design emphasizes the importance of combinatorial approach and plans to eliminate the immunosuppressive tumor burden using Aspirin^®^ to block PGE2 and nivolumab to block PD-1/PD-L1 checkpoint axis.

In 2010, the FDA approved Sipuleucel-T (Provenge, Dendreon), the first therapeutic cancer vaccine. This is an autologous cellular immunotherapy for the treatment of patients with symptomatic or minimally symptomatic metastatic castrate-resistant prostate cancer (CRPC). Sipuleucel-T is composed of autologous APCs cultured ~3 days with a fusion protein that combines recombinant prostatic acid phosphatase (PAP) with recombinant GM-CSF (PAP-GM-CSF). Approval for Sipuleucel-T was based on three pivotal phase III clinical trials (D9901, D9901/D9902A, and IMPACT). The results showed that Sipuleucel-T was well tolerated with an overall survival period of 28.5 months versus 21.7 months in the placebo group, and with a median survival benefit of 4.1 months [113]. In addition to PAP, PSMA was also explored as a target for prostate cancer immunotherapy. Xi et al. published the results of a clinical study in which autologous moDCs pulsed with recombinant PSMA and recombinant Survivin were administrated to 21 patients with CRPC. Results of the study showed that treatment with the DC vaccine prolonged patients’ overall survival by 11 months compared with those treated with docetaxel plus prednisone [114]. In addition, safety, and efficacy of an innovative DC vaccine (mDC, pDC, and the combination of mDC/pDC) loaded with TAA MUC1, NY-ESO-1, and MAGE-C2 was evaluated for the treatment of CRPC patients (NCT02692976).

Tanyi et al. have recently described the immunization of 25 patients with recurrent ovarian cancer using autologous DC vaccine pulsed with autologous oxidized whole-tumor lysate alone, in combination with intravenous bevacizumab, or bevacizumab plus low-dose intravenous cyclophosphamide. They demonstrated that this combination was feasible, safe, and well tolerated, and it elicited antitumor immunity [115]. Considering DC vaccination studies concerning glioblastoma, Liau et al. reported interim results of an ongoing phase III clinical trial evaluating the safety and efficacy of autologous tumor lysate-pulsed dendritic cell vaccine (DCVax^®^-L) in newly diagnosed glioblastoma. The addition of DCVax^®^-L after surgery and chemotherapy was safe and feasible and extended patient survival [116].

## 8. Conclusions

The development of cell-based immunotherapies is both a promising and challenging task for modern transfusion medicine. However, the increasing endeavors of transfusion centers and similar academic institutions in this manner are of great importance for future availability of such state-of-the-art therapeutic approaches for patients. Although it would be unrealistic to expect for academic centers to pursue larger, multicenter trials, their efforts in demonstrating both safety and efficacy in smaller IIT trials is of great value. Furthermore, at least under regulatory directives of the European Union, cellular immunotherapies with demonstrated safety and efficacy profiles could be subsequently implemented in a hospital exemption scenario, where patients could be treated in their local state hospitals much sooner than what it would take for cutting-edge cellular immunotherapies to receive marketing authorization.

## Figures and Tables

**Figure 1 ijms-22-05120-f001:**
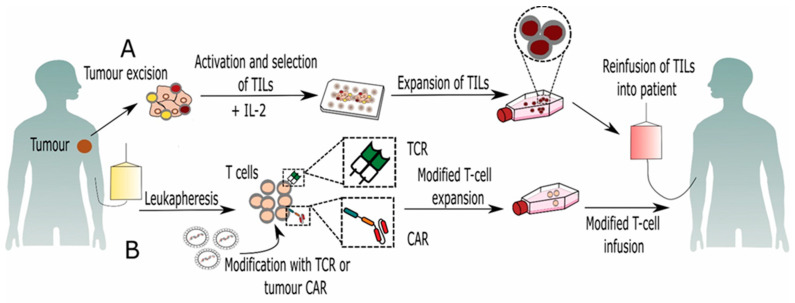
Adoptive T-cell therapy. (**A**) TILs are isolated from excised tumors and expanded with high doses of IL-2. These highly activated TILs are then infused back into the patient following lymphodepleting chemotherapy. (**B**) In TCR/CAR-based ACT, T cells are isolated by leukapheresis and modified to express either TCR or CAR by gene transfection. Modified T cells are expanded ex vivo and infused into the patient.

**Figure 2 ijms-22-05120-f002:**
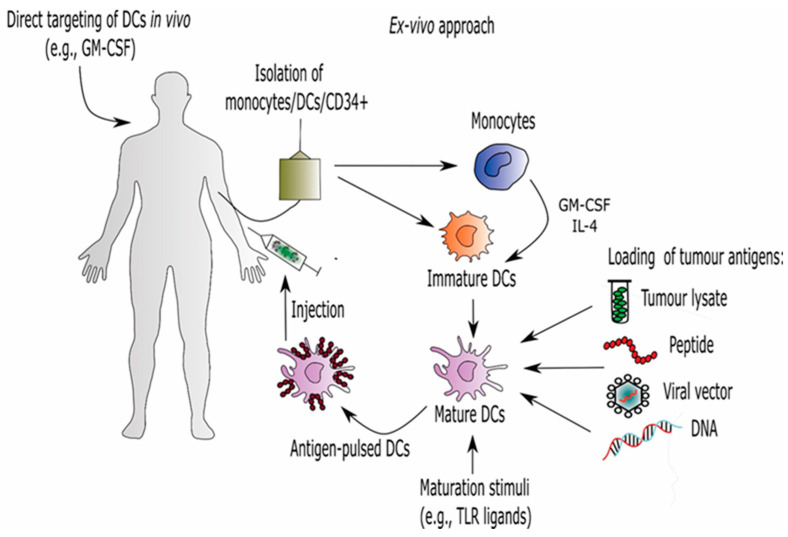
Dendritic cell-based immunotherapy. Approaches for generating DCs-based vaccines include direct expansion of circulating DCs, isolation of circulating DC subsets, and differentiation from monocytes or CD34+ progenitor cells. After differentiation, immature DCs are activated with different maturation stimuli and loaded with tumor antigens. Finally, prepared antigen-loaded DCs are injected to the patient.

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
