# Peer review of "Overview of Cellular Immunotherapies within Transfusion Medicine for the Treatment of Malignant Diseases"

_ijms, 2021, doi:10.3390/ijms22105120_

Round 1
Reviewer 1 Report
This review article authored by Nataša Tešić and colleagues outlined the overall picture of current knowledge about cellular immunotherapies for the treatment of malignant diseases. They started with a solid background by summarizing the development of transfusion medicine, and then described the application of adoptive cell therapy for cancer treatment. In the main analysis section, the authors reviewed the mainstream immune cell-based immunotherapies by fully integrated the primary literature articles. Overall, this is a nicely organized paper with the most up-to-date and comprehensive information of the field.
Suggestions:
- The heading of part 3 is “Genetically modified T lymphocytes”, while only 3.1 and 3.2 belong to this category. So the main analysis section of part 3 needs to be reorganized in logic, especially the “NK cell” and “DC” parts.
- Some abbreviations are not defined before use (“TAAs” in line 135) and some are inconsistent (for example: “GvHD” in line 394 and 395 vs “GVHD” in other places).
Author Response
- Thank you for this correction. We have appropriately corrected and adjusted the headings in a logical fashion.
- We have corrected the missing abbreviations and unified them as requested.
Reviewer 2 Report
This is a comprehensive, well written review on a really hot topic such as cell-based cancer immunotherapy. Surprisingly, out of 110 references, only one was published in 2019, with the rest of them being older. Recent advances in the field, which is moving ahead extremely fast, are therefore missing. For example, four CAR-T cell therapies have been FDA-approved to date, but only the first two are considered in the manuscript. The last one (lisocabtagene maraleucel, Breyanzi) was approved on February 5, 2021, perhaps too recently to be included, although the clinical trial TRANSCEND (NCT02631044) could have been mentioned. However, brexucabtagene autoleucel (Tecartus), approved on July 24, 2020 based on ZUMA-2 (NCT02601313), is also ignored. Results of this clinical trial were reported in N Engl J Med in 2020 (doi: 10.1056/NEJMoa1914347). On the other hand, information about probably less relevant “not yet recruiting” trials is provided.
The authors are kindly requested to review and comment recent literature on the topic, as well as to check the current status of the clinical trials previously included, prioritizing trials with published results and obviously therapies approved by FDA and/or EMA.
Minor comments:
Line 148: “In the past few years, fourth-generation CAR-T cells, so-called “TRUCKs” have been additionally modified with transgenic protein, for instance cytokine IL-12”. The term “transgenic protein” is confusing. Cells are modified with transgenes to express transgenic proteins, being the CAR itself one of them. I would propose "have been additionally modified to express cytokines such as IL-12".
Line 173: “every treatment can result in tissue damage due to the unwanted expression of the transferred gene in normal tissues”. This phrase is difficult to interpret. T cells are genetically modified ex vivo to express CAR, so no gene transference to normal tissue is possible in vivo. Perhaps the authors refer to damage due to expression of the target antigen on normal cells?
Lines 319, 322: K567 is used instead of K562.
Author Response
We thank the reviewer for his sincere comments. We have added several references, as well as certain important registered trials. Many of them are dated from 2020 on. We have also added additional discussion in this context, which is marked in the manuscript (colored yellow). We believe this will raise the quality of the manuscript.
Minor comments:
We have corrected as the reviewer implied. For the second comment - yes, it was meant the target antigen. Thank you.
Reviewer 3 Report
The manuscript is well written, organized, and comprehensive.
Minor concern: A short paragraph related to the emerging T cell-based ACT for the treatment of COVID-19 would make it even more recent and more attractive to the readers.
Author Response
This is indeed an interesting proposal. Although our manuscript is focused on cellular immunotherapy of cancer, we agree with the reviewer and have added a short paragraph in section 4.2. (marked yellow). The increase in ACT therapy studies for Covid-19 is indeed very extensive and worth mentioning.
Round 2
Reviewer 2 Report
Line 122: Clinical trial NCT03158935 is not recruiting. It was completed in August 2020
Lines 334 and 337 (319 and 322 in the previous versión): In references 59 and 60, the authors use K562 cells, K567 cells do not exist. This has not been corrected as suggested.
Line 336: “inhibitory and costimulatory molecules were also transduced”. No inhibitory molecule is used to produce aAPCs in ref. 59, and it makes no sense anyway.
Author Response
We have corrected all three minor points as the reviewer requested. We apologize for the repetitive flaws.